# Brain Substrates for Distinct Spatial Processing Components Contributing to Hemineglect in Humans

**DOI:** 10.3390/brainsci11121584

**Published:** 2021-11-29

**Authors:** Yann Cojan, Arnaud Saj, Patrik Vuilleumier

**Affiliations:** 1Laboratory for Behavioral Neurology and Imaging of Cognition, Neuroscience Department, University of Geneva, 1211 Geneva, Switzerland; arnaud.saj@umontreal.ca (A.S.); patrik.vuilleumier@unige.ch (P.V.); 2CRIR/Institut Nazareth et Louis-Braille du CISSS de la Montérégie-Centre, Longueuil, QC J4K 5G4, Canada; 3Department of Psychology, University of Montréal, Montréal, QC H3A 1G1, Canada; 4Neurology Department, Neuropsychology Unit, University Hospital of Geneva, 1205 Geneva, Switzerland

**Keywords:** attentional processes, functional MRI, parietal lobe, frontal lobe, spatial neglect

## Abstract

Several cortical and sub-cortical regions in the right hemisphere, particularly in parietal and frontal lobe, but also in temporal lobe and thalamus, are part of neural networks critically implicated in spatial and attentional functions. Damage to different sites within these networks can cause hemispatial neglect. The aim of this study was to identify the neural substrates of different spatial processing components that are known to contribute to neglect symptoms. First, three different spatial tasks (visual search, bisection, and visual memory) were tested in 27 patients with focal right brain-damage. Voxel-based lesion–symptom mapping was used to determine the relationships between specific sites of damage and severity of deficits in these three spatial tasks. Secondly, fMRI was used in 26 healthy controls who performed the same tasks. In the healthy group, fMRI results showed a differential activation of regions within the parietal and frontal lobes during bisection and visual search, respectively. In the patients, we confirmed a critical role of right lateral parietal cortex in bisection, but lesions in frontal and temporal lobe were more critical for visual search. These data support the existence of distinct components in spatial attentional processes that might be damaged to different degrees in neglect patients.

## 1. Introduction

The neural mechanisms of spatial attention have been extensively dissected in different models [1,2]. Several neuroimaging studies revealed that brain systems responsible for representing space and directing attention in space implicate a distributed network of cortical and subcortical areas, interconnecting parietal and frontal regions with anterior cingulate cortex, basal ganglia, and thalamus [3,4]. Some areas within this network overlap with neural systems involved in the control of eye movements, including the superior parietal regions in intraparatietal sulcus (IPS) and the frontal eye field (FEF) [5]. Recent results suggest that different attentional functions might be subserved by partly segregated networks [3,6,7] with intraparietal cortex and superior frontal cortex being more important for voluntary (endogenous) attention, and the temporoparietal junction and inferior frontal cortex being more important for reflexive (exogenous) attention to salient or unexpected stimuli.

Damage to these networks (particularly in right hemisphere) is commonly associated to contralesional hemispatial neglect, a severe neuropsychological syndrome characterized by impaired awareness for sensory stimuli located in the side of space opposite to the damaged hemisphere [3,6,7]. While many aspects of this syndrome are attributed to impaired control of spatial attention, the exact mechanisms and neural circuits affected in these patients main unresolved, and clinical deficits may vary across patients with neglect being observed in some tasks and not others. Several studies using lesion-symptom voxel-based mapping have tried to relate these clinical differences to anatomical differences in lesion sites [3,6,7], e.g., relating more explorative components of neglect to frontal regions and more perceptual components to parietal or temporal regions. However, functional brain imaging studies in normal subjects during various spatial tasks usually impaired in neglect patients have pointed to a common role of intraparietal sulcus and middle/superior frontal cortices across several spatial domains, including selective orienting of attention [8,9], visual search [10,11,12], line bisection [13], or spatial working memory [4,8,14].

In this study, we examined more precisely the neural underpinnings of different aspects of spatial cognition, by combining an fMRI paradigm in healthy subjects and statistic voxelwise lesion mapping in brain-damaged patients. By using a comprehensive series of behavioral tests, we delineated neural networks underlying distinct spatial processing domains, including bisection, exploration, and short-term memory.

Our aim was to compare these different spatial tasks in healthy subjects and right brain-damaged patients to identify both common and distinctive neural substrates.

## 2. Experiment 1

### 2.1. Methods

#### 2.1.1. Participants

Twenty-six healthy volunteers were recruited to perform a task specifically designed to probe different attentional and spatial processes in fMRI (see below). The participants were aged 29.6 + 3.44 years (range 25–39), 11 women and 15 men. All participants signed an informed consent according to procedures approved by the local ethical committee.

#### 2.1.2. Behavioral “Triplet Task” Design during fMRI

Our visual paradigm was specifically designed to probe distinct spatial processes, including perceptual, attentional, and memory components, and to keep sensory inputs constant across different task conditions (see Figure 1).

Three tasks using the same visual stimuli were given to participants, each task requiring a similar binary response (indicated by key-press):(i)Bisection task: participants must indicate whether the central item is located at the midpoint between the two others, or not (response: yes or no).(ii)Visual search task: participants must indicate whether the single-odd item in triplets is a square or a diamond (response: square or diamond).(iii)Memory task: participants must indicate whether the three items appeared at the same location as in the preceding trial, or not (response: same vs. different). Successive trials were constrained such that the positions of items on trial *n* corresponded to positions on trial *n* − 1 on 50% of all trials (one-back task), irrespective of the actual shape (square, diamond, triangle) of items.

These different tasks probed distinct neural systems involved in spatial processing, including perceptual, attentional, and working memory functions, while visual inputs were kept constant across all conditions. Each task was given in blocks of 30 trials (10 × 3 triplet positions on the screen), separated by short rest blocks (fixation point only), and repeated in a different order in three different scanning runs. This yielded a total of 30 trials (22 min of experiment) with stimuli presented in left visual field (LVF), right visual field (RVF), or centrally (BVF), for each of the three tasks.

All visual stimuli were back-projected onto a mirror mounted on the MRI head coil, and all responses were recorded by a keypad for subsequent analysis of the participant’s performance. Eye movements were not recorded in the scanner.

#### 2.1.3. Acquisition of fMRI Data

MRI data were acquired in the Biomedical Imaging Center at Geneva University Hospital and Brain and Behavior Laboratory (BBL) at University Medical Center, using a 3T whole-body TRIO system (Siemens) with the standard head-coil configuration. Functional T2*-weighted images were obtained using echoplanar imaging (EPI) with axial slices (TR/TE/Flip = 2200 ms/30 ms/85°, FOV = 235 mm, matrix = 128 × 128). Each functional volume comprised 32 contiguous 3.5 mm-thick slices, parallel to the inferior surface of occipital and temporal lobes, with a final voxel size of 3 × 3 × 3 mm. For each patient, a high-resolution anatomical image was also acquired after the functional scans, using a 3D-GRE T1-weighted sequence (FOV = 250 mm, TR/TE/Flip = 15 ms/5.0 ms/30°, matrix = 256 × 256, slice thickness = 1.25 mm).

#### 2.1.4. Analysis of fMRI Data

All fMRI data were processed and analyzed using the general linear model for event-related designs in SPM8 (Wellcome Department of Imaging Neuroscience, London, UK; http://www.fil.ion.ucl.ac.uk/spm, accessed on 10 November 2021). Functional images were realigned, corrected for slice timing, normalized to an EPI-template (re-sampled at a voxel-size of 3 mm), spatially smoothed (8 mm FWHM), and high-pass filtered (cutoff 180 s). Statistical analyses were performed on a voxelwise basis across the whole-brain, using a mixed blocked and event-related design [15].

Individual visual events were modeled by a standard synthetic hemodynamic response function (HRF) corresponding to 3 stimulus types (triplets presented in LVF, RVF, or centrally) in each of the 3 task conditions (bisection, search, memory), resulting in 9 independent event regressors. To take into account any sustained shifts in baseline activity associated with the different task blocks, we added 4 additional regressors for the 3 task epochs and the rest epochs, modelled by a boxcar waveform convolved with the HRF.

Parameter estimates for each regressor were estimated at each voxel by GLM using a least-square fit to the data, for each condition and each individual participant. Statistical parametric maps of the t-statistic (SPM[t]) generated from linear contrasts between conditions in individual subjects were then included in a second-stage random-effect analysis, using one-sample t-tests on the contrast images obtained from each condition in each subject (Friston et al., 1998). The resulting random-effect maps SPM[T] were thresholded voxelwise at conventional statistical values (*p* < 0.001 uncorrected, with a cluster threshold of *p* < 0.05).

Main comparisons were performed between each task (bisection, search, memory) and rest, and between the different tasks among them. Conjunctions were tested for potential overlap between conditions. Together, these analyses enabled us identifying the neural systems common to different spatial processing domains, as well as those that are selectively responsible for bisection, search, or spatial memory functions.

### 2.2. Results

#### 2.2.1. Behavioral Performance during fMRI Scanning 

Correct Responses. Analysis of errors on all three tasks showed no effect of session (*p* = 0.52) or visual field (*p* = 0.08), and no interaction. There was a significant task effect (F(2, 150) = 21.87, *p* < 0.001). The errors (Figure 2a) were significantly greater for the line bisection task (6.7 ± 1.5%) compared with the other two tasks (Spatial memory: 1.5 ± 0.7%; Visual Search: 2.6 ± 1.1%). In the bisection task, the participants showed more errors for central and right visual field compared to left visual field (7.1 ± 1.5% for right and center vs. 6.2 ± 1.7% for left; *p* = 0.04). In the visual search task, left (2.8 ± 1.1%) and right (3.0 ± 1.3%) visual field showed more errors compared to central visual field (2.0 ± 0.8%, *p* = 0.05 and *p* = 0.04 respectively). For memory task, no difference was observed between visual fields (left: 1.3 ± 0.7%; center: 1.2 ± 0.6%; right: 1.5 ± 0.8%; left vs. center: *p* = 0.84; left vs. right: *p* = 0.13; center vs. right: *p* = 0.20).

Reaction Time. Analysis of reaction time (ms, Figure 2b) on all three tasks showed a significant interaction between Task and Visual field (F(4336) = 7.40; *p* < 0.001). In the bisection task, no difference was observed between visual field (left: 684.3 ± 12.5; center: 679.3 ± 15.5; right: 696.0 ± 13.4). For visual search task, central visual field showed slower RTs compared to left (686.6 ± 15.8 and 653.1 ± 12.6 respectively; *p* = 0.04) and right (656.2 ± 14.6; *p* = 0.04) visual fields. No difference was found between left and right visual fields (*p* = 0.78). For memory task, the central (644.6 ± 14.6) visual field showed faster RTs compared to left (683.1 ± 15.8; *p* = 0.005) and right (684.5 ± 16.5; *p* = 0.006). No differences between left and right visual fields were observed (*p* = 0.78).

#### 2.2.2. Event-Related fMRI Results 

The analysis of brain imaging data (FWE, *p* < 0.005, Figure 3) showed significant activations in the bilaterally parietal cortex, the left motor areas, the right occipital-temporal gyrus and right middle frontal lobe for the three tasks (bisection, visual search, spatial memory).

##### Bisection Task

The contrast Bisection > Visual search + Spatial memory showed a specific activity in the middle occipital gyrus region (Brodmann area 19) extending to superior parietal cortex (Brodmann area 7). The simple contrasts (Bisection > Visual search; Bisection > Spatial memory) revealed a superior parietal lobe (x = 18 y = −61 z = 55, Z = 2.67, *p* = 0.004, Brodmann area 7; Figure 4a) and a middle occipital lobe (x = 33 y = −88 z = 31, Z = 4.16, *p* < 0.001, Brodmann area 19) activity.

##### Visual Search Task

The main contrast Visual search > Bisection + Spatial memory showed bilateral activity in the middle frontal gyrus (left: x = −36 y = 50 z = 1, Z = 3.41, *p* < 0.001; right: x = 33 y = 50 z = 4, Z = 2.92, *p* = 0.002) and in the right post central gyrus (x = 15 y = −55 z = 70, Z = 2.46, *p* = 0.007). The simple contrasts (Visual search > Bisection; Visual search > Spatial memory) confirmed higher activities in the superior frontal lobe (x = 21 y = 5 z = 55, Z = 3.23, *p* = 0.001, Figure 4b) and in the superior parietal lobe (x = 15 y = −64 z = 52, Z = 2.81, *p* = 0.002).

##### Spatial Memory Task

The main contrast Spatial memory > Visual search + Bisection showed significant activity specifically in the left middle temporal gyrus (x = −48 y = 8 z = −35, Z = 3.50, *p* < 0.001, Brodmann area 21), cerebellum (x = −39 y = −43 z = −29, Z = 3.91, *p* < 0.001), and parahippocampal gyrus. The simple contrasts confirmed this involvement of the parahippocampal gyrus in the Spatial task (x = −33 y = −28 z = −17, Z = 3.35, *p* < 0.001). 

In addition, in this fMRI experiment, there was a preponderant involvement of right hemisphere activations for bisection and search tasks. These results are in line with previous findings [16] and our expectations for a right-hemisphere dominance in the spatial tasks used.

Bisection, visual search, and spatial memory are tasks commonly used for testing attentional deficits in patients with unilateral neglect, and in particular the right-lateralized networks mediating attentional functions [8,17,18]. Moreover, they were found to be differentially impaired by lesions located in different brain regions in stroke patients with or without neglect [7,19]. Therefore, an additional study was carried out in brain-damaged patients to determine whether performance in these three tasks was associated with distinct patterns of lesion in regions of interest similar to those identified in controls.

## 3. Experiment 2

### 3.1. Methods

#### 3.1.1. Participants

Twenty-seven patients (Table 1) were recruited among stroke patients admitted to the Neurology Department, University Hospital of Geneva. All patients had a first right-hemisphere stroke, hemorrhagic or ischemic, demonstrated by MRI or CT scan. Spatial neglect [20] and others neuropsychological syndrome were assessed using a standard battery of clinical tests. All patients had normal function in both ears on the basis of their clinical history and neurological examination during the clinical workup. The severity of neglect was assessed by averaging scores from standard tests including the bells cancellation task, Figure Copy, and Line Bisection (see Table 1). All patients were also examined using a routine battery of standardized clinical tests, including mini-mental state examination, to exclude dementia and any other major cognitive disorder that would impact on task performance and collaboration. Patients gave informed consent according to the local ethics rules of the University Hospital of Geneva.

#### 3.1.2. Brain Imaging and Lesion Analysis

For each patient, brain lesions were demonstrated by clinical MRI scans and reconstructed on axial slices using MRIcro [21], according to previously described methods [7,18,22,23,24]. Lesioned areas were transformed to a three-dimensional region of interest (ROI) corresponding to the lesion volume, and normalized to a standard brain template using MRIcro. The normalized lesion ROIs were then superimposed on a T2 MRI template in order to determine side views of lesions (see Figure 5).

The obtained lesions (regions of interest) were then submitted to voxel-based lesion-symptom mapping (VLSM) [7,18,22,23,24] in order to determine the critical brain regions implicated in Bisection task, Visual Search task, or Spatial Memory task. On a voxel-by-voxel basis, the VLSM algorithm separated the subjects into two groups according to the presence (or lack) of a lesion in that voxel. Then, a t-test was performed based on behavioral scores. The resulting statistical measure was used to create a separate map based on each variable. Areas showing significant correlations with behavioral disorders were identified using the false discovery rate corrected at *p* < 0.01.

#### 3.1.3. Behavioral “Triplet Task” 

Patients were comfortable sited in front of the computer screen on which the ’triplet task’ visual paradigm was presented (see section Behavioral “triplet task” design during fMRI of Experiment 1).

### 3.2. Results

#### 3.2.1. Behavioral Performance

Correct Responses. Analysis of errors on all three tasks showed no main effect of session (*p* = 0.07), no visual field effect (*p* = 0.50), and no interaction. The data showed only a significant task effect (F(2, 156) = 84.92, *p* < 0.001). The errors (Figure 6a) were significantly greater for the line bisection (15.5 ± 1.5%) and search tasks (15.2 ± 1.1%) compared with the spatial memory task (6.7 ± 0.7%). In the bisection task, the participants showed more errors in left (16.4 ± 1.4%) visual field compared with central (15.3 ± 1.6%; *p* = 0.09) and right (14.8 ± 1.7%; *p* = 0.04) visual field. In the visual search task (left: 15.2 ± 1.1%; center: 15.3 ± 0.6%; right: 15.0 ± 1.3%; left vs. center: *p* = 0.99; left vs. right: *p* = 0.68; center vs. right: *p* = 0.95) and spatial memory task (left: 6.4 ± 0.7%; center: 6.3 ± 0.6%; right: 7.3 ± 0.8%; left vs. center: *p* = 0.86; left vs. right: *p* = 0.10; center vs. right: *p* = 0.17), no differences were observed between the visual field conditions.

Response Time. Analysis of response time (ms, Figure 6b) on all three tasks showed a significant interaction between Task and Visual field (F(4304) = 26.65; *p* < 0.001). In the bisection task, the left (1011.4 ± 29.1 ms) visual field produced significant slower reaction times compared to the two other visual field conditions (center: 943.6 ± 25.4; right: 957.2 ± 25.1; left vs. center: *p* = 0.01; left vs. right: *p* = 0.03; center vs. right: *p* = 0.64). In the visual search task, the left (1060.7 ± 31.5 ms) visual field showed a significant difference compared to the two other visual fields (center: 993.5 ± 28.7; right: 833.0 ± 29.9; left vs. center: *p* = 0.009; left vs. right: *p* = 0.021; center vs. right: *p* = 0.64). For memory task, no difference was observed between the visual fields (left: 905.7 ± 20.4; center: 874.7 ± 19.9%; right: 921.8 ± 23.3%; left vs. center: *p* = 0.17; left vs. right: *p* = 0.48; center vs. right: *p* = 0.09).

#### 3.2.2. VLSM Analysis

The global VLSM (including the error score of three task, *p* < 0.005, Figure 7) showed significant areas of lesions in the right occipital-temporal gyrus and the right parietal cortex, with peak in supramarginal gyrus (x = 38, y = −48, z = 27) for the three tasks (bisection, visual search, spatial memory).

##### Bisection Task

Poor performance on the bisection task was associated with larger lesions in the right posterior hemisphere (Figure 7), including the angular gyrus and the inferior parietal cortex (peak MNI coordinate x = 40, y = −51, z = 28, *p* < 0.01).

##### Visual Search Task

The VLSM analysis showed that lesions in middle frontal gyrus (x = 28, y = 19, z = 42) and supramarginal gyrus (x = 39, y = −48, z = 27) were associated with deficits in performance of visual search in brain-damaged patients (Figure 7).

##### Spatial Memory Task

The deficits in spatial memory (Figure 7) were associated with the hippocampus lesion (x = 32, y = −32, z = −9), as well as supramarginal gyrus (x = 37, y = −47, z = 25).

## 4. Discussion

We used the bisection, visual search, and spatial memory tasks to identify both common and specific regions subserving different spatial cognitive functions within the attentional network. All three tasks required voluntary attention and activated the dorsal attentional system, with a specific right hemisphere role. 

### 4.1. Bisection Task

In the present study, we found fMRI activations mainly in right superior parietal lobe and middle occipital lobe, but weaker effects in middle frontal gyrus for the bisection task. Those regions were already reported for the landmark task or line bisection task [13,25]. Remarkably, we also found predominant damage to posterior parieto-occipital areas associated with deficits in our bisection task in stroke patients.

Our results are consistent with other neuroimaging studies in healthy subjects. Indeed recent work [25] compared the cortical activation in both variants of ’bisection’: a landmark task and a manual line bisection task. During the landmark task, the authors found an activation of right IPS, anterior cingulate gyrus, and right lateral peristriate cortex [13,26], whereas during the manual line bisection, activations were located in right IPS, right FEF and right lateral peristriate cortex.

In an event-related potentials study, Fink’s team [27] showed that the main areas involved in a perceptual bisection tasks were right middle occipital gyrus, right superior posterior parietal cortex, bilateral inferior occipital gyrus, and right inferior posterior parietal cortex. The areas are well known as critical epicenters within the neurocognitive network that subserves spatial attention and mental visuo-spatial representations.

A recruitment of these brain areas may accord with their role in the representation of egocentric and allocentric space as described in several previous studies [18,28,29,30]. Converging evidence from these studies indicates that egocentric spatial coding relies on a network of frontoparietal areas including superior parietal lobule, intraparietal sulcus and precuneus, the temporoparietal junction, as well as premotor, inferior frontal cortex, and cerebellum. Among these, the superior parietal lobule and precuneus appeared particularly important for egocentric space coding and were found to strongly activate when a subject is asked to locate body parts [7]. Indeed, damage to these regions may be responsible for deficits in the perception [30] and representation of one’s own body [18]. Bisection performance may also recruit neural circuits for allocentric spatial coding, mediated by object processing and navigation systems in occipital and temporal lobes as observed here in addition to parietal cortex, and ultimately integrated with attentional systems in dorsal networks.

Lesion studies in patients with brain damage also confirmed the involvement of these structures in spatial cognition. Recent work in our team [7,18] on large series of brain-damaged patients has suggested a dissociation in the attentional network. Indeed using a voxelwise mapping technique, Verdon et al. (2010) were able to identify specific brain regions for different attentional components affected in neglect patients. In this study, a large number of patients (80) were evaluated in a battery of standard neuropsychological tests, and individual scores from a factor analysis based on these tests pointed to three main components of spatial neglect (perception/visual-spatial; exploration/visual-motor, and allocentric/object-centered). The perception component was located specifically in the right inferior parietal lobe; the exploration component in the right dorsolateral prefrontal cortex; and the object-centered component in the temporal lobe. Our bisection task which was associated with the first factor (perception), confirmed that the inferior parietal lobe is critical for visuospatial abilities underlying this task, according to both normal subjects with fMRI data and patients with lesions in this region, but also consistently with data from the literature on spatial neglect [17].

### 4.2. Visual Search Task

Our visual search task mainly activated the middle frontal gyrus, but also the superior parietal lobe. This accords previous results, which showed an involvement of middle frontal gyrus in the visual search task [31,32].

Similar fronto-parietal activities have also been also observed in a recent study using the Embedded Figures Test [33]. During the latter, activity in parietal and frontal areas, in the right hemisphere was correlated with behavioral performances during the search task. The authors proposed that these posterior parietal and frontal regions are necessary in the selective processing of visuo-spatial information in healthy subjects, with inhibition of distracting information. Likewise, the factor analysis performed by Verdon et al. (2010) showed that frontal cortex is mainly loading on a visuomotor component of neglect and necessary for visual search in cancellation tasks [7]. Although the involvement of the frontal lobe might be distinctively related to spatial functions recruited by visual search, it is worth noting the concomitant involvement of the parietal cortex for this task. This region might make other specific contributions to the visual search task, including the spatial organization of search paths and working memory for searched locations. These findings are also consistent with the literature of spatial neglect, suggesting that patients with frontal lesions may be more affected on visual search tasks [34].

### 4.3. Attentional Network

Our results confirm the idea that spatial deficits in patients depend on the lesion site and affect different tasks to different degrees, and more generally that parts of inferior parietal lobe and dorsolateral frontal lobe within the attentional network are critically associated with spatial neglect. Those results also indirectly reinforce the hypothesis of a disconnection in white matter pathways connecting these two regions that could lead to combinations of similar deficits [6,7,23,35].

A common finding in the literature is that patients with neglect typically have larger lesions and a greater number of different areas damaged as compared with non-neglect patients (e.g., [36,37,38]. The patients with persistent neglect also show more frequent involvement of white matter tracts in the paraventricular region, possibly involving branches of the superior longitudinal fascicle (SLF) connecting parietal and frontal cortex [38,39]. Therefore, damage to these white-matter tracts has been suspected to play a critical role in neglect [40]. This disconnection hypothesis is further supported by the effect of intra-operative electrical stimulation directly applied to the SLF [35], showing that disruption of the second segment of the SLF can produce significant deviation during line bisection. Likewise, Ptak et al. (2020) suggested that damage to the right cerebral hemisphere affects spatial cognition through impaired functional connectivity between particular cortical nodes. They also highlighted the importance of interhemispheric connections for the manifestation of spatial deficits after focal damage to the right hemisphere, with a central role of the right temporo-parietal junction and frontoparietal connections for contralateral deployment of attention and detection of task-relevant stimuli [41].

Although more research is needed to clarify the role of different white-matter pathways in sustaining communication between different brain areas during spatial processing tasks [42], these data support the notion that neglect may result from a combination of deficits reflecting different spatial component processes, which become disrupted after damage to extensive parts of the fronto-parietal attention network or damage to multiple connections within this network [6,7].

In contrast, we found that our spatial memory task was not reliably associated with well-delineated neural substrates in the fMRI study, and only weak and restricted effects of lesions in the temporal lobe. This is consistent with the functional anatomy of memory systems in temporal lobe and their relative sparing in neglect syndrome.

## 5. Conclusions

We find common activations in brain regions for different spatial tasks, overlapping with the proposed dorsal attention network around IPS, FEF, and lateral occipital cortex [6]. In addition, we also show a distinctive role for two main regions in frontal and parieto-occipital areas for bisection and search tasks, respectively, suggesting an antero-posterior segregation of functions within the attentional network [6], which may be defective in different ways in patients with spatial neglect with different sites or extents of lesions.

## Figures and Tables

**Figure 1 brainsci-11-01584-f001:**
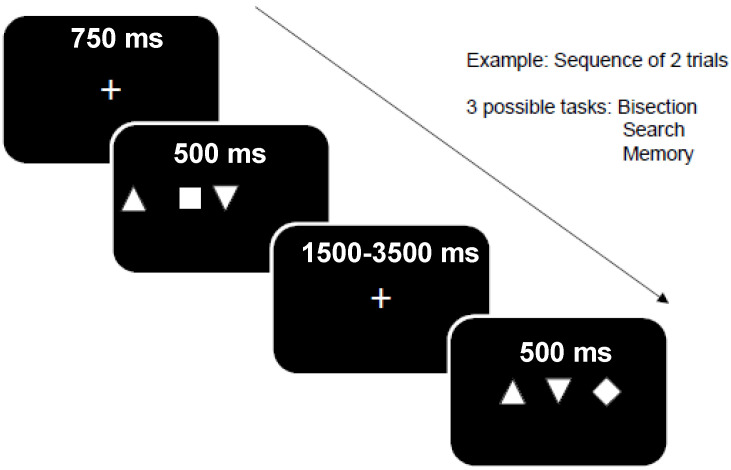
The “triplet” visual task used the during fMRI experiment. The same stimuli and trial sequences were used for three possible tasks: bisection, search, and memory.

**Figure 2 brainsci-11-01584-f002:**
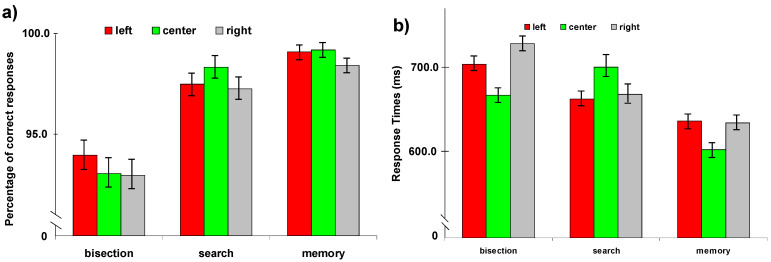
(**a**) Accuracy (% correct) and (**b**) responses time (ms) are depicted as a function of the triplet stimulus position in left- (red), centre (green), and right visual field (grey) in healthy controls subjects.

**Figure 3 brainsci-11-01584-f003:**
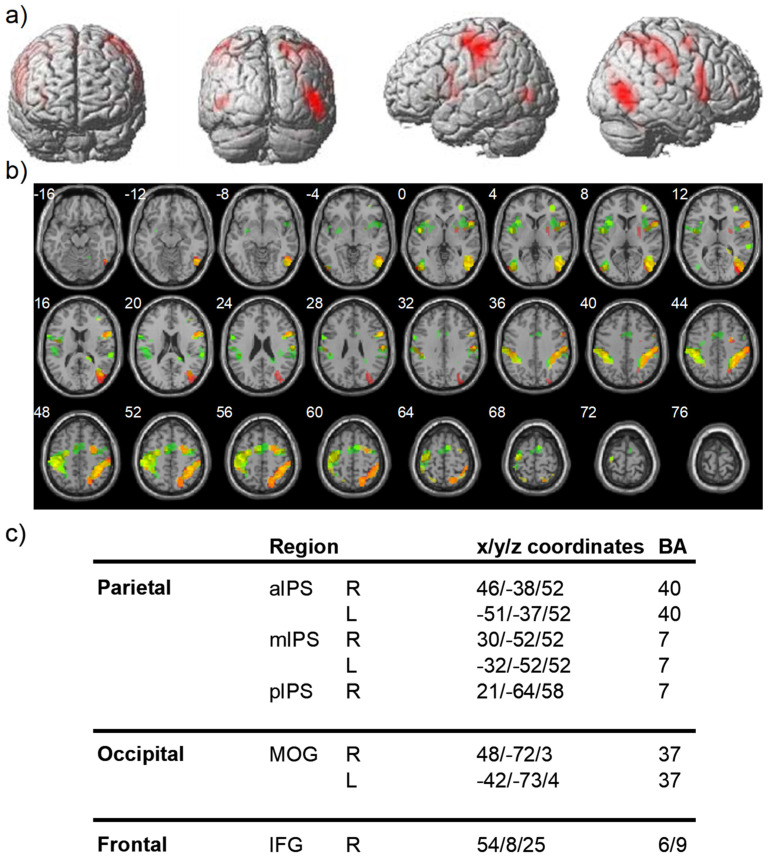
Common areas activated by all tasks. (**a**) 3D rendering of a standardized T1 brain template with superimposed loci of brain which are commonly activated by bisection, visual search, and spatial memory tasks. (**b**) Axial slices of a standardized brain template showing overlapping regions of activity for bisection (red), visual search (yellow), and spatial memory (green). (**c**) Coordinates of activation maxima for regions of overlap and corresponding Brodmann’s Areas (BA). aIPS: anterior Intra Parietal Sulcus, mIPS: medial Intra Parietal Sulcus, plPS: postero Intra Parietal Sulcus, MOG: middle occipital gyrus, IFG: Inferior Frontal Gyrus, R: right, L:Left.

**Figure 4 brainsci-11-01584-f004:**
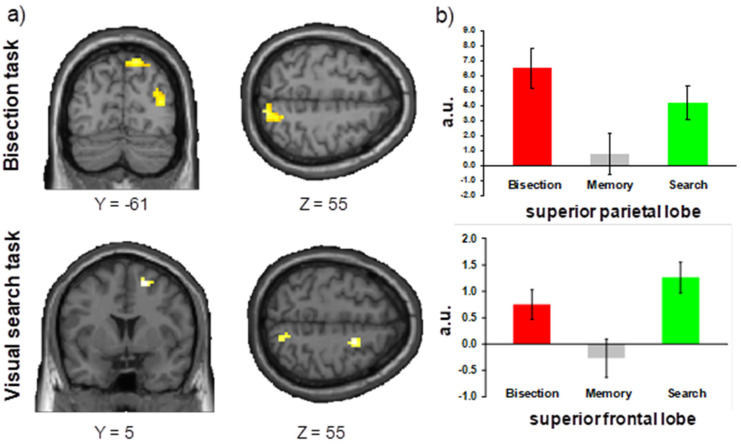
(**a**) Region of interest analyses for bisection task and visual search task. Brain regions activated in our main whole-brain contrasts are projected on standard anatomical template. (**b**) Parameter estimates of activity (beta value, in arbitrary units, averaged across responsive voxels in each cluster) are shown for the bisection tasks and visual search in two activated regions of interest, IPS and MFG. Red colors = bisection, grey colors = spatial memory, and green colors = visual search.

**Figure 5 brainsci-11-01584-f005:**
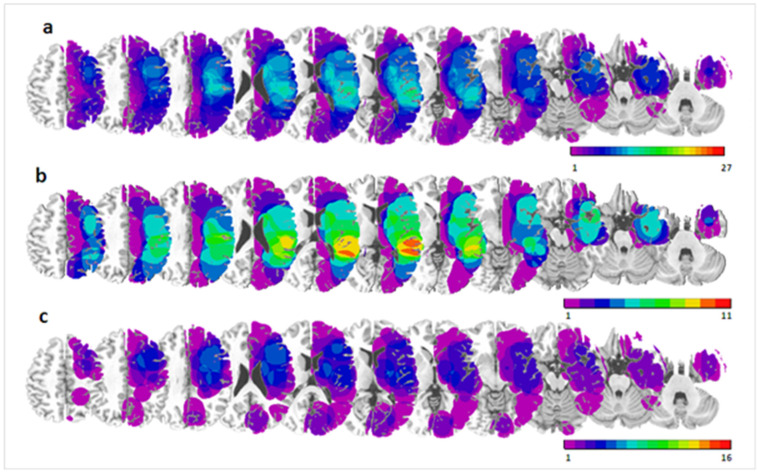
Overlap of the brain lesions for (**a**) all 27 patients included in our study, (**b**) a subgroup of 11 patients showing consistent neglect in all clinical tests, and (**c**) a subgroup of 16 patients showing no neglect in any of the clinical test.

**Figure 6 brainsci-11-01584-f006:**
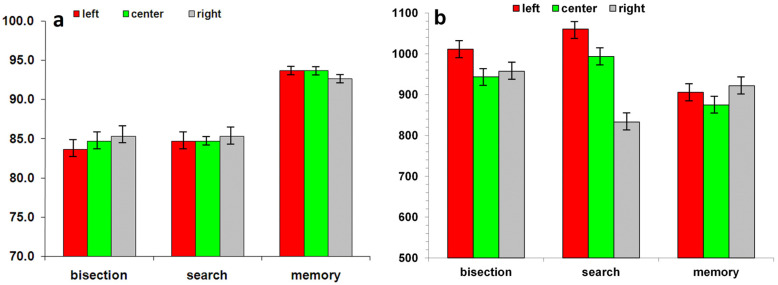
(**a**) Accuracy (% correct) and (**b**) responses time (ms) are depicted as a function of the triplet stimulus position in left- (red), center (green), and right visual field (grey) in right brain-damaged patients.

**Figure 7 brainsci-11-01584-f007:**
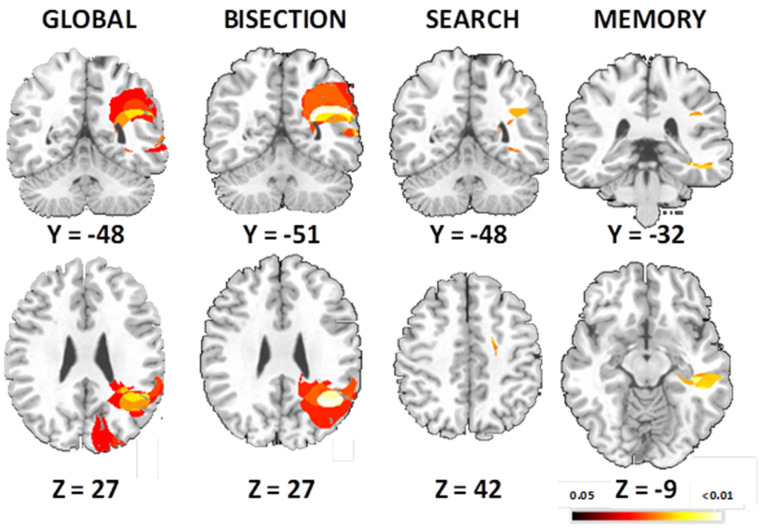
Voxel-wise lesion symptom maps show all voxels surviving a 1% false discovery rate cut-off threshold. Lesion-symptom maps for all right brain-damaged patients showing for the three tasks, bisection task, visual search task and spatial memory task.

**Table 1 brainsci-11-01584-t001:** Demographic and clinical data of patients.

	Spatial Neglect	No Neglect
Age	73.1 (12.1)	64 (6.1)
Sex (F/M)	7/4	6/10
Etiology (isch/hem)	9/2	14/2
Mini Mental State (/27)	24.9 (1.9)	25.2 (1.0)
Bell cancellation (total omission)	17.5 (4.6)	3.0 (2.3)
Line bisection (% deviation)	47.7 (20.9)	4.7 (4.2)
Copy of scene (omission)	1.8 (1.5)	0.1 (0.2)

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
