# Peer review of "Brain Substrates for Distinct Spatial Processing Components Contributing to Hemineglect in Humans"

_brainsci, 2021, doi:10.3390/brainsci11121584_

Round 1
Reviewer 1 Report
Summary. The aim of this work is to clarify which areas of the fronto-parietal network support some visuo-spatial tasks that are also used in the diagnosis of spatial eminegligence. In this regard, both "healthy" participants and patients with right hemisphere damage performed three tasks: a line bisection task, a visual search task, and a visual-spatial memory task. The "healthy" patients performed the tasks during an fMRI. Results from the two groups of participants showed that the superior and inferior parietal lobule (i.e., angular and supramarginal gyrus) were more involved in line bisection, whereas the middle frontal gyrus was more involved in the visual search task. Finally, the spatial memory task was more supported by hippocampal/parahippocampal structures. These results would indicate that an antero-posterior brain network is involved in attentional processes and that this network is defective in patients with spatial neglect.
General comment. The work is very interesting and carefully conducted. For these reasons it could represent a useful contribution to the literature. However, the work needs more revisions before it can be accepted for publication.
Specific comments.
Abstract: The structure of the abstract does not correspond to that of the manuscript. In fact, in the manuscript, data on "healthy" participants are presented first and then on patients contrary to what is stated in the abstract.
Introduction: The introduction does not adequately explain the rationale of the study, thus failing to appreciate the novelty of this work compared to other similar ones conducted in the past. Moreover, the hypotheses are missing. Given the enormous amount of work on the subject, it would seem unconvincing to pass this work off as merely "exploratory". Therefore please enrich the introduction so that more specific hypotheses/expected results can be constructed.
Discussion: as with the introduction, the discussion should be further developed. Specifically, the authors could consider the distinction between egocentric and allocentric reference systems (Galati et al., 2010) in line bisection tasks. Such a distinction is also crucial for patients with neglect (Leyland et al., 2017). Indeed, the tasks proposed by the authors all require an allocentric reference system, whereas the results found indicate areas usually involved in the use of egocentric reference systems. For a neural link between dorsal attentional systems and egocentric reference system consider the study by Ruotolo and colleagues (2019, Neuroscience). Further reflection on this point is requested from the authors.
In addition, results from spatial memory tasks are no longer discussed. I request the authors to elaborate on this part as well.
Author Response
Thank you for your positive comments and constructive suggestions. In our revision, we have addressed all the suggestions as described below.

Reviewer 2 Report
This is a well written, interesting paper that investigates brain regions involved in tasks affected by hemispatial neglect and secondly voxel morphometry in patients with hemispatial neglect . A lot of excellent work has gone into analyzing the regions involved in the tasks as well as analyzing patient data. The paper is very straightforward, however it could use more description. Specifically:
1-How can the two analyses be connected. For instance, in healthy individuals, the bisection task activates the parietal cortex, and VLSM analysis implicates lesions in the parietal cortex as well. Do these regions overlap?
2- There is little discussion of the spatial memory task. I realize the authors were trying to keep the stimuli identical across tasks, but this task may not truly recruit areas involved in memory. There is strong evidences that visual memory tasks recruit the intraparietal sulcus . Specifically, Sheremata, Bettencourt, & Somers showed that this area is involved while holding visual stimuli constant. It is therefore surprising that these areas were not recruited in the memory task described here. A description of why this discrepancy might have occurred is necessary.
In addition some details about the experiment would be helpful.
1- For instance, the VLSM analysis is never explained (indeed the abbreviation is never spelled out).
2- Can you include the exact p-values in line 248?
3- Please add standard deviation to behavioral results (error and reaction time).
4- Please add error bars in Figure 2 and Figure 6.
Author Response
We thank the reviewer for his/her positive assessment and are grateful for the constructive comments. We detail below how we attempted to clarify the issues you raised.

Round 2
Reviewer 1 Report
I am satisfied with the revisions to the manuscript made by the authors